# Shear Thickening Fluid and Its Application in Impact Protection: A Review

**DOI:** 10.3390/polym15102238

**Published:** 2023-05-09

**Authors:** Haiqing Liu, Kunkun Fu, Xiaoyu Cui, Huixin Zhu, Bin Yang

**Affiliations:** School of Aerospace Engineering and Applied Mechanics, Tongji University, Shanghai 200092, China

**Keywords:** shear thickening fluids, soft body armors, dampers, shock absorbers, multi-function

## Abstract

Shear thickening fluid (STF) is a dense colloidal suspension of nanoparticles in a carrier fluid in which the viscosity increases dramatically with a rise in shear rate. Due to the excellent energy absorption and energy dissipation of STF, there is a desire to employ STFs in a variety of impact applications. In this study, a comprehensive review on STFs’ applications is presented. First, several common shear thickening mechanisms are discussed in this paper. The applications of different STF impregnated fabric composites and the STF’s contributions on improving the impact, ballistic and stab resistance performance have also been presented. Moreover, recent developments of STF’s applications, including dampers and shock absorbers, are included in this review. In addition, some novel applications (acoustic structure, STF-TENG and electrospun nonwoven mats) based on STF are summarized, to suggest the challenges of future research and propose some more deterministic research directions, e.g., potential trends for applications of STF.

## 1. Introduction

Shear thickening fluid (STF) is a non-Newtonian fluid that exhibits an abrupt increase in viscosity by a few orders of magnitude with increasing shear rate [1,2,3]. STF behaves as a solid-like material under applied stress due to increasing viscosity, and when the loading is removed from the medium, the STF turns to the initial liquid state. Due to the excellent energy absorption and energy dissipation characteristics of STF, it has been widely used in the Li-ion batteries [4], wearable devices [5], triboelectric nanogenerator (TENG) [6], protective structures [7,8] and some novel applications [9,10,11].

STF consists of a carrier liquid and colloidal particles [12,13]. The particles are generally selected from a number of groups of particles which include silica, polymethyl methacrylate, calcium carbonate, cornstarch, synthetically and naturally occurring minerals, polymers or a mixture of them. Many carrier fluids such as water, ethylene glycol (EG) and poly ethylene glycol (PEG) have been investigated [14]. The common particles and carrier fluids of STFs are summarized in Table 1. The shear thickening mechanisms, material composition, fabrication methods, rheological properties and factors influencing shear thickening behavior have been reviewed by many researchers [15,16,17,18,19,20,21].

According to the rheological properties of STF, they can be divided into two categories: continuous shear thickening (CST) and discontinuous shear thickening (DST). CST is observed below threshold value, $\emptyset_{c}$, and becomes weaker with decreasing volume fraction. DST is an abrupt increase by orders of magnitude in viscosity above a critical value of the applied shear rate [44]. Fernandez et al. [45] proposed a model to identify the nature of the Shear thickening transition which was controlled by the volume fraction and boundary lubrication friction coefficient through the simulations and experiments. Brown et al. [46] gave a good overview on phenomenology and mechanisms of shear thickening and discussed the relations to jamming systems. They proposed different mechanisms and models to explain the common physical properties and a phase diagram for shear thickening behavior. The rheological properties of STF are affected by many factors, including particle volume fraction, particle aspect ratio, particle–particle interactions, hardness, roughness, particle size, size distribution of particle, modification of particle, liquid medium, pH value, temperature and additives of STF, as reviewed by Gürgen [47] and Subramaniam [48]. More recently, Yusuf Salim [49] has reviewed the factors and damage mechanism of the stab and spike resistance performance of STF-impregnated. To improve the impact resistance of textile, different STFs were prepared for the high performance fabric composites [50]. Some approaches to improve the ballistic performance including surface treatments and modifications of fabrics in STF-based soft body armor were elucidated [11,51]. The applications of various STF impregnated fabric composites and the STFs’ contribution on improving the impact, ballistic and stab resistance performance were investigated by many researchers [21,52]. Mechanistic problems due to sudden change in viscosity and recent developments of simulations of the effect of contact forces were discussed in the STF [53]. The premise of this paper is to review the recent developments in various applications of STF, as shown in Figure 1.

The shear thickening mechanisms will be discussed, including the order–disorder theory, hydro-clustering theory, dilation theory, jamming theory and friction contact theory. Recent advances, including soft body armors, dampers, shock absorbers and some novel applications, such as sensors, aerospace, acoustics, battery in the engineering of STF are summarized in this review. Finally, the challenges of future research and the prospects of STF are also discussed.

## 2. Shear Thickening Mechanism

Over the past few decades, the shear thickening mechanism has been studied by many researchers and extensive studies have been reported in the literature, including the order–disorder transition, hydro-clustering, dilation, jamming and friction contact theory.

### 2.1. Order–Disorder Transition Theory

Hoffman [38] first proposed the order–disorder transition theory that the shear thickening phenomenon was concurrent with the transition from order to less ordered flow of particles. Different diffraction patterns were found before and after shear thickening, as shown in Figure 2a. Subsequently, he found that the nanoparticles in the STF are layered order below the critical shear rate and the transition from order to disorder caused a drastic increase in suspension viscosity by experiment [60]. At a certain critical shear rate, particle doublets would form, which moved out of their layers and disrupted the flow, resulting in an increase in the suspension viscosity [61]. Moreover, Laun et al. [62] studied shear-induced particle structures of STF with styrene-ethylacrylate-copolymer spheres in glycol or water by small angle neutron scattering in a wide range of shear rates and found that the particle structures which stopped flowing depended on the shear rate. In an effort to better understand the shear thickening mechanism, computer simulations presented a path forward. Catherall et al. [63] used Stokesian dynamics to investigate the rheology and microstructure of STF under the shear thickening regime; they controlled the interparticle gaps to evaluate the thickening behavior, reporting that at higher hard core volume fractions, larger jumps in viscosity were observed with the transition from order to disorder and the strong thickening behavior was only observed with the enhanced lubricating force.

### 2.2. Hydro-Clustering Theory

Brady et al. [66] and Butera et al. [67] found that the order–disorder transition did not always occur during all shear thickening phenomena, suggesting that the order–disorder transition was dispensable for shear thickening. Moreover, Bossis et al. [39] also found that shear thickening behavior was dependent on the formation of large clusters rather than the order–disorder transition. The hydro-clustering mechanism proposed that the particles of STF were driven together into clusters under shear, as a result of short-range hydro-dynamic lubrication forces overcoming the repulsive forces among adjacent particles. The hydro-clustering mechanism has been widely accepted for explaining the shear thickening behavior by many researchers [68,69]. For example, Cheng et al. [64] visualized and identified the hydro-clusters as the onset of thickening in the STF of silica spheres in a water-glycerin mixture using the fast confocal microscopy with simultaneous force measurements. Figure 2b depicts the instantaneous real-space configuration of hydro-clusters and different colors indicate different clusters. Next, Maranzano and colleagues [70] demonstrated the extreme sensitivity of high-shear rheology to the surface properties of suspended particles, which was consistent with the formation of hydro-clusters and the dominance of short-range lubrication forces in the shear thickening state. Later, Chellamuthu et al. [71] measured the extensional properties of fumed silica nanoparticles in polypropylene glycol as a function of concentration and extension rate and found that the dynamic rheological behavior of STF was caused by the formation of large hydro-dynamic clusters. Brady et al. [72] used the Stokesian dynamics to calculate the particle trajectories to find that molecular-dynamics-like method could accurately represent the suspension hydrodynamics.

### 2.3. Dilation Theory

Brown et al. suggested that the hydro-clustering theory, which had been successful for clarifying CST, had failed to explain the orders of magnitude increase in viscosity during DST [28,36]. In this mechanism, the shear stress overcomes the onset stresses of the shear thickening, and they begin to shear relative to each other, which causes the grain packing to dilate. Dilation of granular shear flows causes particles to penetrate the liquid-air interface for STF, generating restorative forces transmitted through the suspensions produce a confining shear stress which is proportional to normal stress, resulting in DST [28]. Figure 2c depicts the visible effect of dilation on STF of cornstarch in water [28]. Shear thickening was described as dilatancy, which refers to the expansion of a system due to the change in packing arrangement [73,74]. A dense mixture of granules and liquid is called a dilatant fluid which often exhibits shear thickening behavior with the increase in shear rate. Nakanishi et al. [75] constructed a fluid dynamics model for the dilatant fluid by introducing a phenomenological state variable for a local state of nanoparticles, and the results of model showed that the STF exhibited an instability in a shear flow and shear thickening oscillation.

### 2.4. Jamming Theory

In recent years, studies revealed that the explanation of the DST was closely connected to jamming [76,77]. They believed that the nanoparticles of STF would spontaneously aggregate under shear, forming local blockages. The dispersed phase particles diffuse under the action of shearing, and the suspension cannot flow at all, showing the exponential growth of shear stress and the explosive increase in apparent viscosity. Next, a new added mass model was introduced to clarify the dynamic solidification process that the large normal stresses formed using a rapidly growing jammed solid region, which is pushed through the surrounding STF by the impactor [78]. Then, Marc-Andre Brassard et al. [79] reported a new model, in which the viscous-like forces control the impact response of STFs to supplement the add-mass theory. Peters et al. [65] explored the solid behavior in the shear-jammed regime experimentally by dropping small steel spheres (diameter 5.0 mm) onto the STF. The dynamic shear jamming behavior was observed directly, as shown in Figure 2d. Moreover, Seto et al. [80] adopted a numerical method which included hydrodynamics interactions and granular contacts, and observed that contact friction was essential for discontinuous shear thickening. A low viscosity occurred in a contactless (hence, frictionless) state, and a high viscosity exhibited a frictional shear-jammed state. Next, the elongation and breakage of a filament of STF under tensile loading was closely related to the jamming transition seen in its shear rheology as presented by Smith et al. [81]. The jamming theory can well explain the solid–liquid transition in STF, it provides a reasonable explanation for the relationship between shear stress, shear rate and apparent viscosity. However, jamming is just a general term for describing the phenomenon and does not really reveal the nature of the shear thickening phenomenon.

### 2.5. Friction Contact Theory

The friction contact theory was adopted to explain the relationship between the CST and DST. Under low shear, the normal contact force between particles is small, and the fluid lubricating force plays a more important role in the impact process. When the normal contact force between particles is large, the fluid film between particles is destroyed, and particle-to-particle contact force and friction force play a leading role. As the shear rate increases, there are more frictional contacts, and the system forms a frictional contact network [45]. Mari et al. [44] numerically obtained that as the shear rate increased, the shear increased as shown in Figure 2e. The uncontacted particles with gray colored line, which connects the centers of two related particles, are drawn, while the red line indicates the particles that will come into contact during the thickening process. Clearly, the contacting particles form an extended contact network in the STF. All the aforementioned studies show that the mutual frictional contact of dispersed phase particles plays an important role in the shear thickening process of STF. In summary, the friction contact model is a theoretical model accepted widely by scholars. It can not only explain the CST behavior and the DST behavior at the same time, but also can be verified through inverse shear rheological experiments and numerical simulations.

## 3. Soft Body Armors

Soft body armor is an advanced protective equipment which is used to protect the human body against attacks of various kinds of sharp objects or projectiles [54]. When a high-performance fabric is hit by a projectile, energy is absorbed through various mechanisms, depending on both material and projectile parameters. The research on the preparation of high-performance STF-impregnated fabrics for soft body protection has become a research hotspot. Firstly, the fabrics were cut into a square to prepare the specimens of STF-impregnated fabrics. It is difficult to make the uniform impregnation of fabrics with a high viscosity suspension. Therefore, the STF was diluted in ethanol with a volume ratio, and the fabrics were soaked in the solution under an ultrasonic treatment. After that, the roller shown in Figure 3a [35] was used to squeeze the excess solution, and the STF-impregnated fabrics were dried in a vacuum oven [18]. Currently, soft body armor research is focusing on the development of light weight, flexible and comfortable armors with improved yarn pull-out behavior, ballistic impact resistance and stab, spike resistance of STF/fabric composites [82].

### 3.1. Inter-Yarn Interaction of High Performance STF/Fabric Composites

Novel body armor based on STF has shown promising prospects towards improved protection and flexibility [83,84,85,86]. The improvement in fiber friction has been proved to be a major contribution of STFs to impregnated fabrics. Mawkhlieng et al. [35] explored the role of STF in enhancing the impact resistance of high-performance Kevlar fabrics. They found that the inherent shear thickening behavior of silica-PEG STF played a crucial role other than just increasing the yarn-to-yarn friction of the Kevlar fabrics. In addition, the study by Sanchi Arora et al. [87] implied the interplay between ultra-high molecular weight polyethylene (UHMWPE) woven fabrics with 400 denier and STFs of silica particles in water and PEG-200, as shown in Figure 3b. For firm structures created by higher values of fabric set, STF-treated fabric/firm structures deteriorated the impact resistance owing to stress concentration. For fabrics woven with finer yarns, the yarn-to-yarn friction was enhanced after STF impregnation.

Yarn pull-out test is a good way to understand the role of friction in the performance of fabrics [88]. Figure 3c depicts the basic principle of the single yarn pullout test. The STF impregnated fabrics improved the yarn pull-out force which was found to show significant correlation with the energy absorption of Kevlar and UHMWPE (Spectra) fabrics during low-velocity impact (6 ms^−1^) [89]. In the yarn pull-out test conducted by Bai et al. [90], an energy absorption model was adopted to investigate the energy absorption mechanism that the work conducted by external force can be equivalent to the interfacial friction energy, and they found that the energy absorption capacity of STF-treated fabric was obviously affected by the SiO_2_ mass fraction of STF and the yarn pullout speed was much greater than that of neat fabric. Srivastava et al. [26] investigated the effect of padding pressure on the impact energy absorption and measured the yarn to yarn friction, which only partially influenced the impact performance of Kevlar woven fabrics impregnated with silica/PEG STF by the quasi-static yarn pull-out force.

In consideration of the requirement that the body-amour materials should be applied under various environments, Wang et al. [41] studied the shear thickening behaviors of STF containing polystyrene (PS) microspheres in PEG-200 at different temperatures. The 55 wt% STF represented the best shear thickening behavior at −15 °C. Figure 3d illustrates the inter-yarns frictional forces of STFs-treated fabrics at different pull-out speeds. The results show that the lower temperature improves the thickening effect, and the frictional force presents a more important role than hydrodynamics in the aggregation for these shear thickening fluids. Some textile fabric materials were impregnated with different STFs to study the effect of the STF treatments on the impact response and inter-yarn friction of fabrics by researchers, as listed in Table 2.

**Table 2 polymers-15-02238-t002:** STF composition and fabric materials used to study the effect of the treatments on the impact response and inter-yarn friction of fabrics.

Particles	Carrier Fluids	Additives	Fabrics Materials	Reference
Silica nanoparticles	PEG	—	Kevlar fabrics	[84]
Silica nanoparticles	PEG	Polyvinyl alcohol (PVA)	Kevlar fabrics	[26]
Silica nanoparticles	PEG	—	UHMWPE fabrics	[89]
Silica nanoparticles	EG	—	Kevlar fabrics	[90]
PS microspheres	PEG	—	Kevlar fabrics	[41]
Silica nanoparticles	PEG	Silicon carbide	Twaron	[32]

**Figure 3 polymers-15-02238-f003:**
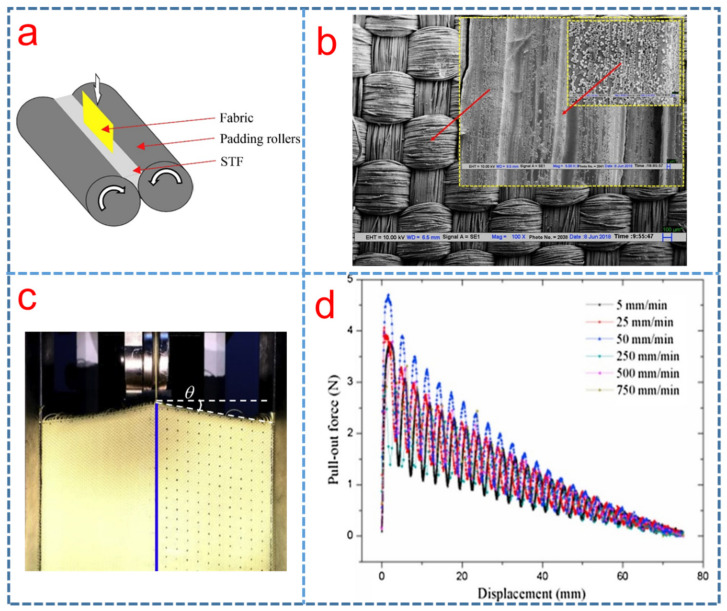
(**a**) STF impregnated fabric paddling process [35], (**b**) SEM image of STF treated fabric woven with 50 × 50 EPI × PPI from 400 denier yarn [87], (**c**) photos of yarn pullout test before tension and after tension [89] and (**d**) pull-out force versus displacement for STF-treated Kevlar fabrics [41].

### 3.2. Application of STF/Fabric Composites in Ballistic Protection

Lu et al. [91] investigated the ballistic behavior of STF-treated fabrics under high velocity impact as shown in Figure 4a, and found that the movement constraint of the primary yarns decreased the projectile velocity. Figure 4b shows the failure characteristics of post-impact fabrics [92]. The failure mode of STF-impregnated fabric was changed from tensile-dominant to shear-dominant and the pull-out distance was decreased, causing a decrease in energy absorption. Moreover, the ballistic performance of Twaron^®^ CT615 plain-woven fabric treated with different silica colloidal particle water suspension (SWS) was studied by V.B.C. Tan [27]. It was found that the ballistic limits of fabric armor systems could be improved by impregnating the fabric with SWS due to the rise in projectile-fabric friction and inter-yarn friction arising from the silica particle addition and silica clusters formation. Wagner et al. [93] also described the use of STFs with different volume fractions to improve the ballistic resistance of the Kevlar fabric.

Ávila et al. [42] investigated the dual-phase STF of a combination of nano-silica and calcium carbonate with PEG and ethanol and their experimental results showed that the STF consisting of 25% *w*/*w* nano-silica and 75% *w*/*w* calcium showed the best ballistic resistance due to the increase in the inter-yarn fraction. In addition, Gürgen et al. [94] fabricated the multi-phase STFs consisting of silica and PEG suspensions with different amount of silicon carbide additives. They found that the multi-phase STFs enhanced the ballistic performance of fabrics and energy absorption in comparison to single-phase STFs. Subsequently, a numerical model was introduced to investigate the impact behavior of the single and multi-phase STF treated textiles. According to the numerical results, there was a good match with the experimental results between target deformations and projectile residual velocities, which yielded a correlation index of 0.9691, as shown in Figure 4c [95]. Meanwhile, Bajya et al. [96] investigated the effect of nano-silica particle size on the ballistic resistance of soft body armor panels against the small arms ammunition. It was found that soft armor panels consisting of fabrics impregnated with STF based on silica particles (500 nm diameter) yielded lower back face signature than the panels impregnated with 100 nm silica-based STF. Moreover, Khodadadi et al. [92] also found that the impact resistance performance of Kevlar fabric was significantly increased with the weight fraction. Edison et al. [97] investigated the ballistic impact behavior of laminated hybrid panel consisted of aluminum alloy, epoxy and Kevlar fabrics treated with STF in Figure 4d, finding that the energy absorption of the composites improved with the STF addition. Meantime, Subhajit Sen et al. [98] numerically studied the ballistic response of the two STF composite structures, i.e., Kevlar-STF-Kevlar sandwich composite and STF impregnated Kevlar and reported the effect of STF on the process of improving ballistic response.

**Figure 4 polymers-15-02238-f004:**
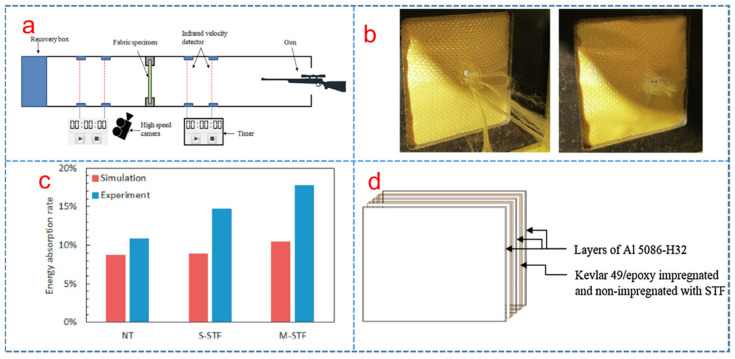
(**a**) The STF-impregnated fabric ballistic test setup schematic diagram [91], (**b**) neat fabric and STF-impregnated fabric damage after ballistic tests [92], (**c**) energy absorption rate of neat and STF-impregnated fabric simulation and experiment [95] and (**d**) hybrid laminate made of Al5086-H32 aluminum alloy, epoxy, and STF-impregnated Kevlar fabric [97].

### 3.3. Application of STF/Fabric in Stab Resistance and Low-Velocity Impact Protection

Figure 5a shows the schematic illustration of the stab resistance of STF impregnated fabric under the impact of a knife, and it was found that the stab resistance of the Kevlar fabrics was significantly improved with an increased addition of STF [99]. Decker et al. [100] investigated the stab resistance of Kevlar and Nylon fabrics impregnated with STFs based on silica particles dispersed in PEG. The puncture resistance increased dramatically under low impact velocity due to the reduced mobility of the yarns. Moreover, Sun et al. [101] reported the stab resistance of an advanced material made up of STF and UHMWPE and found that the great improvement in stab resistant property of the STF/UHMWPE fabric was attributed to the increase in the mass fraction of silica in STF, and that the flexibility of the composite material was higher than that of the neat fabric. Similarly, Xu et al. [102] investigated the effects of silica nanoparticle sizes and silica nanoparticle weight fraction on stabbing resistance of the STF impregnated woven fabric panels. The results indicated that the higher nanoparticle weight fraction and larger nano-particle size of silica resulted in a better stabbing resistance performance. Balali et al. [29] investigated the penetration resistance of glass fiber-reinforced hybrid STF and found that the increase in friction between the penetrator and the fiber, and between the fibers and the yarns at their crossing points enhanced significantly the penetration resistance. Meanwhile, the neat UHMWPE specimen and STF/UHMWPE specimen after the dynamic stab test for knife and spike threats were investigated. It was found that the STF/UHMWPE specimen exhibited less deformation and damage compared with the neat fabrics in both tests [103].

The stab resistance of STF impregnated fabric is affected by many factors, including the particle size [23], hardness [32] and additives [31,34,98,99]. Various kinds of textile fabric materials that were impregnated with STF to enhance their stab and spike resistance were listed by researchers, as shown in Table 3. Meanwhile, Figure 5c is the failure patterns of neat Kevlar and STF-impregnated fabric. Liu et al. [8] investigated a high-impact resistant hybrid sandwich panel filled with STF and found that the energy absorption capacity was effectively improved due to the synergistic effect of STF’s high energy absorption, the improved stiffness of STF-Kevlar fabric and the confinement of STF-filling aluminum cells.

In addition to the above researches on STF impregnated fabric composites soft body armor as a protection solution against impact, bulletproofing and stab, the application of STF on space shields also attracted more attention under the high-speed environment of micrometeoroid and orbital debris (M/OD) [55,104]. Cwalina et al. [55] reported a novel method to improve the cut and puncture resistance of the thermal micrometeoroid garment (TMG) by replacing the standard neoprene-coated nylon absorber layers with woven aramid textiles intercalated with STF, i.e., STF-Armor ^TM^. At equal areal densities, the results showed that a TMG lay-up containing STF-Armor™ greatly improved puncture resistance and reduced total weight with comparable flexibility, as shown in Figure 5c. The study revealed experimentally that the fabric rear wall of a space shielding system could be enhanced effectively by using STF-treated layers [105]. Liu et al. [5] designed a novel carbon nanotubes (CNT)/STF/Kevlar-based wearable electronic textile (ET) composite which had excellent protective and sensing performance for human bodies in different environment and found that it could be applied as a sensor to monitor the signal of various human body movements, as shown in Figure 5d.

**Figure 5 polymers-15-02238-f005:**
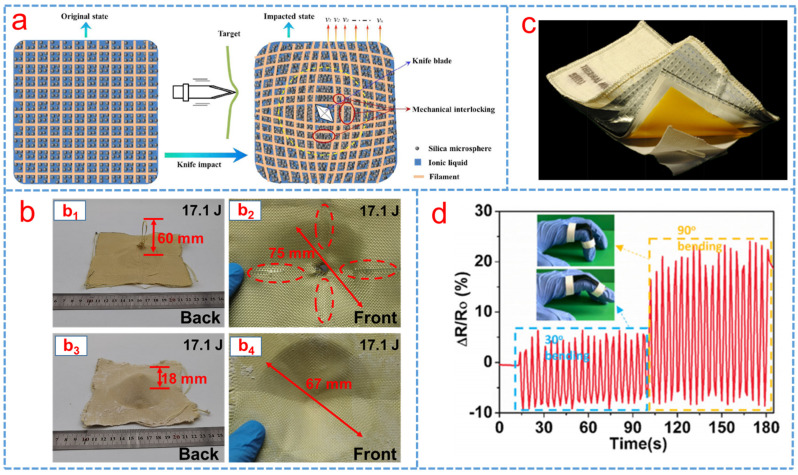
(**a**) Schematic illustration of the stab resistance of the STF−impregnated fabric [99], (**b**) failure patterns of (**b_1_**) and (**b_2_**) neat Kevlar and (**b_3_**) and (**b_4_**) STF−impregnated fabric [8], (**c**) image of the thermal micrometeoroid garment (TMG) [55] and (**d**) relative normalized resistance change in the ET in monitoring movements of a finger [5].

**Table 3 polymers-15-02238-t003:** STF composition and fabric materials to study stab resistance of STF impregnated fabric.

Particles	Carrier Fluids	Additives	Fabrics Materials	Reference
PSt-EA nanospheres	EG	—	Kevlar fabrics	[83]
Silica nanoparticles	PEG	—	Nylon fabrics	[100]
Nanosilica	EG	—	UHMWPE	[101]
Nanosilica	PEG	—	Twaron woven fabrics	[90]
Fumed silica	PEG	Nanoclay	Glass fabrics	[29]
Nanosilica	EG	PEG	UHMWPE	[103]
Fumed silica	PEG	—	Kevlar fabrics	[100]
Kaolin particles	Glycerol	—	Kevlar fabrics	[43]
PMMA	PEG	—	Kevlar fabrics	[23]
Nanosilica	PEG	Silicon carbide	Twaron	[104]
Silica microsphere	[BMIm][BF_4_]	—	Kevlar fabrics	[99]
Fumed silica	PEG	Carbon nanotubes (CNTs)	Woven high modulus polypropylene (HMPP) fabric	[34]
Nanosilica	Ethanol and polyethylene glycol	Silane coupling agent	Kevlar fabrics	[31]

## 4. Dampers

Damping is the most effective method to reduce unwanted vibrations where the system is excited close to its natural frequency. STF-filled dampers have potential application in the industry [106]. The dynamic performance and mechanical model of a self-adaptive STF were investigated and the results showed that the smart damper could realize controllable output damping force by changing the loading frequency, loading amplitude and fluid gap [107]. Neagu et al. [108] studied the micromechanics and damping properties of composites integrated with STFs and found that the stiffness and damping properties were significantly dependent on both frequency and applied external load amplitude. Over the past few decades, extensive studies have been performed to understand the dynamic properties of the shear thickening viscous damper and the results showed that the dynamic properties also affected by the fluid viscoelastic properties [109,110] and particle concentration [111].

Jolly et al. [112] reported the controllable devices consisting of STF that exhibit discontinuous increases in flow resistance as controlled by changes in the applied magnetic or electrical fields. The magnetic-field-controlled and speed-activated magnetorheological STF (MRSTF) consisting of nano-size silica particles in the EG was fabricated [113]. Tian et al. [114] investigated the performance of a linear damper working with MRSTFs and the results showed that the system can investigate the stiffness coefficient and a variable damping coefficient. Furthermore, novel MRSTF-based linear dampers containing 20% and 80% weight fraction micro-sized carbonyl iron particles were investigated [115]. The dampers filled with 20% MRSTF had better magnetorheological effect and shear thickening effect than the 80% magnetorheological STF-filled damper. Moreover, the nonlinear hysteretic behavior and energy dissipation capacity of a STF damper were investigated. According to the responses of damping force-displacement and damping force-velocity, it was found that the loading conditions, regardless of frequency or amplitude, had significant effect on the hysteretic loop and energy dissipation capacity [56]. Gaines reported a STF vibration damper system for vehicle seat to reduce and eliminate vibrations with the improvement in the vehicle seat back [116].

Smart structures based on STFs were also studied for the vibration attenuation and the industrial applications. For example, Gürgen et al. [117] investigated the smart polymer integrated cork composites for enhanced vibration damping properties, finding that STF could contribute to suppressing the vibration in Figure 6a. Subsequently, he reported a novel concept by filling STF into extruded polystyrene foam core of an aluminum face sheet sandwich structure to study the vibration attenuation. From the results, we can come to the conclusion that STFs significantly improved the vibration attenuation of the sandwich structures in Figure 6b [118]. Haris et al. investigated the shock wave mitigation capability of polyurea and STF-based suspension pads, as shown in Figure 6c. The results showed that the STF pad and STF-infused foam pad performed better than the conventional foam pad in terms of peak pressure [119].

Williams et al. [120] proposed surgical and medical garments and materials incorporating STFs, such as surgical gowns, surgical gloves and wound-care products. The use of foam-based STF composites in trauma pads was also investigated. The results of these tests were used to identify the potential usage of STFs in blunt trauma impact resistance applications. Chavhan et al. [57] proposed the STF speed breaker in Figure 6d and found that the STF speed breaker improved the fuel efficiency of vehicles and reduced installation costs and maintenance costs compared to conventional speed breakers. A method to reduce movement of controlled pulse or high energy fracture was reported, where the STF was used as a tamp to minimize damage to the downhole equipment. Subsequently, Wasserman et al. reported a cable consisting of a conductor surrounded by STF systems to resist damage from a puncture by shovels, trucks and other equipment [121].

**Figure 6 polymers-15-02238-f006:**
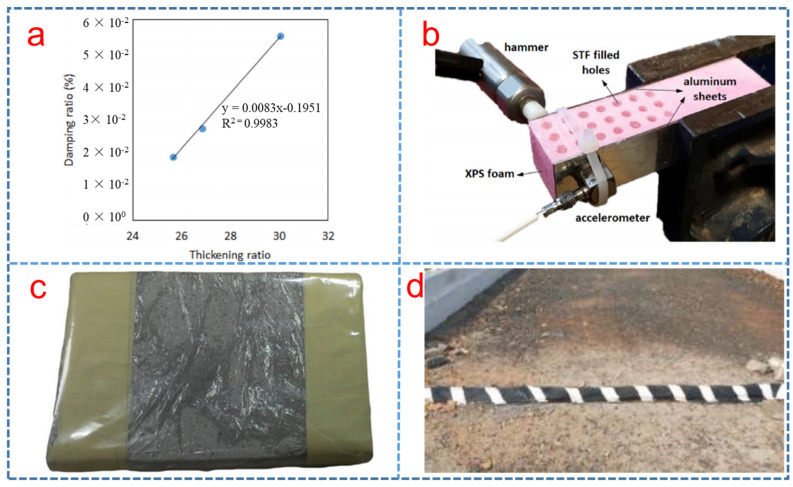
(**a**) Relationship between the damping ratio and STF amount [117] (**b**) Vibration attenuation test of sandwich structures filled STF [118], (**c**) STF-infused foam pad [119] and (**d**) Shear thickening fluid speed breaker [57].

## 5. Shock Absorbers

The shock absorber is a mechanical device, which links the equipment and the foundation, to smooth out, damp shock impulse and dissipate shock energy [122]. STF can be utilized in an impact-resistant structure or energy dissipation attributed to the high stiffness of formation of jamming triggered at the critical shear rate [123]. Because of the following performance of energy absorption and dissipation of STF under compress and impact, it can be used in shock absorber.

The confined compressive behavior of STFs has attracted the considerable interest of many researchers [123,124,125,126,127]. The compression of STFs can be reversible, as shown in Figure 7a. The bulk modulus of STFs was measured and the results showed that the bulk modulus enhanced with the rise in the applied stress [127].

Some studies have used the SHPB to investigate the dynamic compressive behavior under high stain rates [128,129], transient response [130] and stress pules attenuation [131] of STF. Fu et al. [132] investigated the compressive behavior of STF with styrene/arylate particles at 58% volume fraction at high strains rates using the split Hopkinson pressure bar (SHPB) test. The results showed that the impact toughness and energy absorbed increased with the loading rate. In addition, the Johnson-Cook model was used to reproduce the high-rate compressive behavior of STF. Wu et al. [133] investigated the dynamic energy absorption behavior of lattice material filled with STF by the modified SHPB and found that the dynamic energy absorption behavior of the sandwich panel with the STF filled pyramidal lattice truss core could be interpreted by the interaction of the strong lateral drag force between the filled STF and the pyramidal lattice core. The shear thickening behavior of STF was improved with the addition of graphene because the SHPB experimental results showed that the peak flow stress of the STF increased with the increase in the graphene volume fraction [30].

The ballistic response of STF with cornstarch, silicon carbide and silicon dioxide particles was investigated systemically by the experiment [134]. The penetration of STF by a projectile was shown in Figure 7b. The results showed that the nanoparticles have sufficient strength to decrease the projectile velocity, suggesting that the strength of the solid material in the interparticle contacts was overcome by the impact-generated stresses [134].

Brassard et al. [79] found that the viscous-like forces controlled the impact response of STF to supplement the add-mass theory, as shown in Figure 7c. Moreover, Cheng et al. investigated the effect of striker shape on the energy absorption of STF. The results showed that the total impact energy absorption increased with the increase in striker diameter and decrease in the lower penetration depth in Figure 7d [135]. Later, to explore the energy absorption behavior of STF at various temperatures, low velocity impact tests were utilized by a drop-weight tower with a temperature-control chamber and it was found that the anti-impact behavior of the STF enhanced with the decrease in temperature [136]. The restricted boundary conditions played an important role in the impact resistance and energy absorption of STF. It was found that the STF under the finite space constraint of circular section had better impact resistance and energy absorption, as shown in Figure 7e [137].

**Figure 7 polymers-15-02238-f007:**
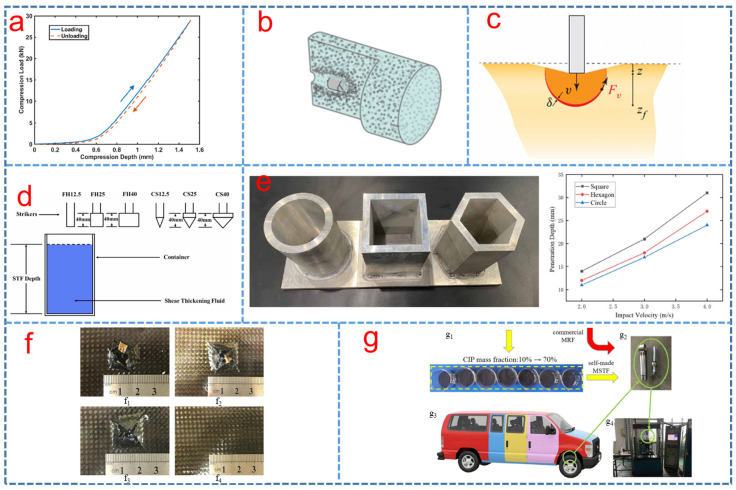
(**a**) Typical compression load and depth of the STF during loading and unloading in a confined compression test [127], (**b**) a broken-out section view of the projectile penetrating the suspension and the related mesostructural changes to the particle distribution [134], (**c**) a front that grows downward and laterally and can experience added-mass and viscous forces [79], (**d**) schematic of low-velocity impact tests with six different strikers [135], (**e**) relationship curves between the drop hammer velocity and the penetration depth obtained by drop hammer impact on STF under finite space constraints of different cross sections [137], (**f**) damage patterns of (**f_1_**) pristine SCP (v_initial_ = 4.0 m/s, t_STF thickeness_ = 7.2 mm), (**f_2_**) STF-filled SCP (v_initial_ = 4.0 m/s, t_STF thickeness_ = 7.2 mm), (**f_3_**) STF-filled SCP (v_initial_ = 2 m/s, t_STF thickeness_ = 7.2 mm) and (**f_4_**) STF-filled SCP (v_initial_ = 2.0 m/s, t_STF thickeness_ = 12.7 mm) [17] and (**g**) MSTF prepared with different CIP mass fractions (**g_1_**), shock absorber used in the test (**g_2_**), the vehicle model (**g_3_**) and test system (**g_4_**) [19].

STF was incorporated with lattice materials or structures to improve dynamic energy absorption by the interaction of lattice structure and STF [123]. The low-velocity impact behaviors of sandwich composite panels (SCPs) with carbon fiber reinforced plastic (CFRP) facings based on a concentrated styrene/acrylate particle STF were investigated by Fu. Figure 7f shows the damage pattern at the back of SCP and STF-filled SCP after impacts at different velocities. The low-velocity impact tests showed that the STF could absorb more energy with less penetration depth than an aluminum foam [17]. The damping behavior and impact properties of the carbon fibers reinforced polymer (CFRP) laminates consisting of STF were investigated, and the results showed that the system can absorb up to 45% of the energy during the impact event at 2.5 m/s due to the STF which can act as shock absorber media [138]. Moreover, Galindo-Rosales et al. [139] reported CorkSTF*µ*fluidics which were eco-friendly light-weight composites comprising of a laminar sheet of compacted micro-agglomerated cork engraved with a network of microchannels by laser and filled with STF of suspension of cornstarch. The results of low-velocity impact tests illustrated that the composites had better energy absorption properties. Furthermore, Liu et al. reported that a multifunctional smart material with both shear thickening effect and magnetorheological performance was fabricated by dispersing carbonyl iron powder (CIP) particles into STF in Figure 7g. The influence of the shear thickening effect of MSTF on the damping force in the shock absorber was studied by testing the self-made MSTF and MRF with the CIP fraction from 10% to 70%. This work provided a design idea to improve the shock absorber performance [19].

## 6. Multi-Functional Properties

Recently, Liu [4] studied the potential of STF in new generation gelled or solid electrolytes to improve the impact resistance of Li-ion batteries. To further understand the response of the electrolyte under external impact, impact experiments were conducted at different speeds, and the results showed that the impact resistant electrolyte has excellent electrochemical stability. The shear thickening effect on electrolytes of lithium-ion batteries was investigated, and the results showed that STFs were proved to be a potential replacement for traditional electrolytes in lithium-ion batteries. The STF could act as both highly conductive electrolytes and mechanical protectors for lithium-ion batteries and could demonstrate shear thickening effect under impact [140]. Wei et al. [141] theoretically studied the vibration of a sandwich beam which consisted of a STF core and conductive skins. The results revealed that the natural frequency of the sandwich beam integrating STF was more dynamic than conventional structures during the different periodic excitations. More recently, the mechanical, acoustic, and thermal performances of STF-filled rigid polyurethane (PU) foam composites were investigated by Li et al. [142]. The STF/PU foam composites increased the compressive, bending strength and maximum acoustic coefficient and were excellent sound-absorbing energy conservation materials. Figure 8a presents the thermal conductivity of rigid STF/PU foam composites of different particle size silica particles. The 1.0 wt% STF of 14 nm silica has the best thermal insulation [142]. As shown in Figure 8b, Liu et al. [7] investigated the acoustic property of 3D printed structures filled with STFs. It was found that sound transmission loss of the structures filled with 46.5 vol% silica-based and 58.8 vol% styrene/acrylate-based STFs have been improved significantly, while their sound absorption coefficient reduced greatly. According the study by Li et al. [143], the sound insulation performance of the STF-treated fabrics was better than that of the untreated fabrics. Their sound insulation was significantly increased by increasing surface density, as shown in Figure 8c. Later, Wang et al. [144] fabricated a novel sound insulation composite by modifying glass fiber fabric with tetrapod ZnO whisker (TW-ZnO)/SiO_2_-compounded STF and came to the same conclusion. Aslan et al. [58] designed an acoustic structure and investigated its sound absorption behavior. The results showed that the frequency at the maximal SAC value decreased with the increase in fluid viscosity, as shown in Figure 8d.

Mechanical components could be fabricated from solid materials containing STFs as discrete droplets or regions or as a co-continuous network within the solid material. Smart components could be fabricated where the stiffness or hardness of a flexible component can change as a result of degree of deformation [145]. Tian et al. [146] reported a rotational brake working with shear thickening fluid, which did not require a power supply due to the changing of torque. The 3D structure of the rotational brake was drawn by the 3D designing software. There was a holder as a reservoir for STF, and the rotator was located on the top of the holder with a ramp fin which was immersed and rotated in STF.

Li et al. proposed a shear thickening polishing method and adopted a material removal rate model to improve rapidly the surface quality, as shown in Figure 9a [147,148,149]. Polishing tests were conducted with a CSM Tribometer by adjusting the equipment for STF polishing. A nano-sized fumed silica-based STF was utilized as a polishing matrix to improve the surface of a steel bar by manipulating the viscosity [59]. Electroosmosis of non-Newtonian fluids was suggested conceptually for pumping, solute transport and heat transport [150]. Xie et al. reported that the STFs were widely applied in enhanced oil technique, adopting a multiphase Lattice Boltzmann method (LBM) model [151]. Recently, the shear dependent electrical property of conductive STFs was investigated [152]. A high anti-impact STF/Ecoflex composite structure with a sensing capacity for wearable design was designed to obtain excellent anti-impact and high energy absorption properties. The results showed that the C-STF/Ecoflex was fabricated by adding carbon nanotubes (CNTs) to STF, which had excellent impact sensing function and could be combined with Kevlar for wearable devices with anti-impact properties due to the STF concentration, as displayed in Figure 9b [153]. A versatile TENG with impact/sway/magnetic field multi-mode energy harvesting and protective properties was developed by assembling shear thickening fluid (STF) and magneto-sensitive films. This novel TENG had been proven to effectively absorb and dissipate collision energy, providing excellent protective property for wearers. Recently, electrospinning was used to fabricate core-sheath fibers and encapsulate STFs in the resultant fibers. A new composite consisting of electrospun ultrafine fibers and STF was developed to improve the shape stability of STF-impregnated fabrics. The composites were demonstrated to be shape-stable with high breakthrough pressure due to the small effective pore size and the high capillary force of UFF membranes [154]. Chen et al. proposed a direct microencapsulation of Ionic-Liquid-based STF and designed a circuit made of Ionic-Liquid-based STF microcapsules-incorporated conductor. The results showed that the circuit can exhibit electrical stability after the impact and autonomic conductivity self-healing after damage [9].

## 7. Conclusions

This paper provides a comprehensive review of various applications pertaining to STFs. There is a growing demand to understand the shear thickening mechanism to explain the mechanical behavior of STFs in different applications. Thus, this review discusses five different shear thickening mechanisms, i.e., order–disorder theory, hydro-clustering theory, dilation theory, jamming theory and friction contact theory. Then, this paper reviews typical STF applications to provide an insight into the usage of STFs in impact protections. This paper reviews different methods to improve the impact, ballistic and stab-resistant performance of STF-impregnated fabric. The greater impact resistance and energy absorption of STF-impregnated fabric are mainly attributed to the inherent shear thickening behavior of STF and improvement in the frictions between yarns and fibers. Moreover, the industrial devices based on STFs, such as dampers, shock absorbers and some other novel applications (acoustic structure, STF-TENG and electrospun nonwoven mats), are also included in this review. With the addition of STF, the mechanical, acoustic, thermal, vibration, mechanic-magnetic coupling and electrokinetic performances of devices are improved.

At present, the application of STF to smart wearable devices has become a hot topic of research. However, the sedimentation of STFs may restrict its wide application. The particle sedimentation of STFs is nearly unavoidable after long-term usage. Therefore, it is recommended to prepare the STF with particles and carrier liquid with comparable densities. Moreover, the shear thickening behavior of STF is also sensitive to the environmental temperature. Preparing an STF that can work under a wide temperature range is a meaningful research direction. However, the problems of high viscosity, hygroscopicity and difficult handling have become the biggest factors hindering its development in the practical application of STFs. To overcome these shortcomings and improve the stability of the STF in service, it is particularly important to use suitable technologies to encapsulate STFs. Hence, the interaction between STFs and different encapsulated materials for impact protection is a potential field for future work.

## Figures and Tables

**Figure 1 polymers-15-02238-f001:**
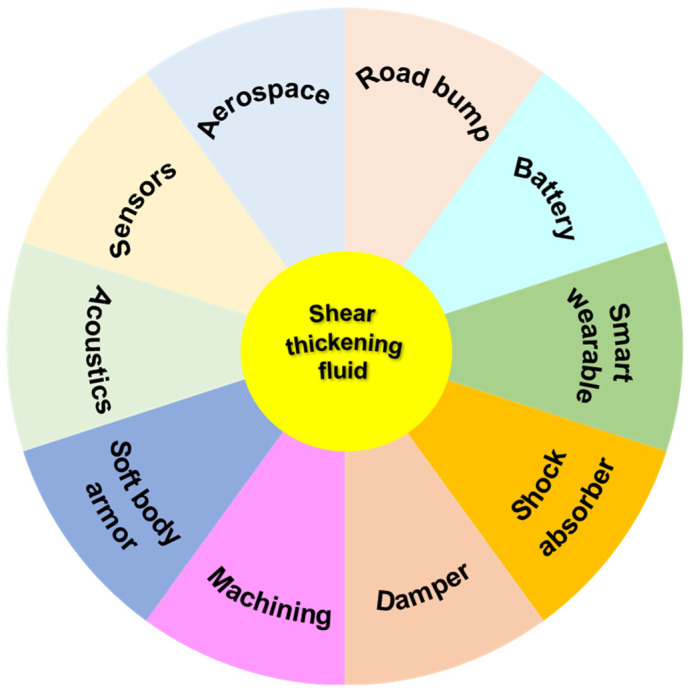
Applications of STF [4,5,6,19,54,55,56,57,58,59].

**Figure 2 polymers-15-02238-f002:**
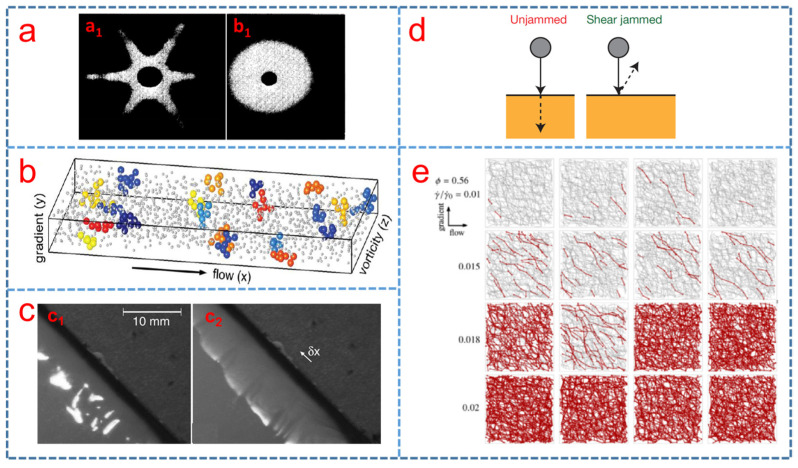
(**a**) Different diffraction patterns before (**a_1_**) and after (**a_2_**) shear thickening [38], (**b**) instantaneous real-space configuration of hydro-clusters [64], (**c**) the visible effect of dilation for shear thickening fluid of cornstarch in water ((**c_1_**) before shear, the surface of the suspension looked wet and shiny, (**c_2_**) when the shear rate was above the critical shear rate, the nearby suspension appeared rough) [28], (**d**) the shear-jammed regime explicitly caused by dropping small steel spheres onto the shear thickening fluid [65] and (**e**) contact networks along the suspension with particle loading of 56% [44].

**Figure 8 polymers-15-02238-f008:**
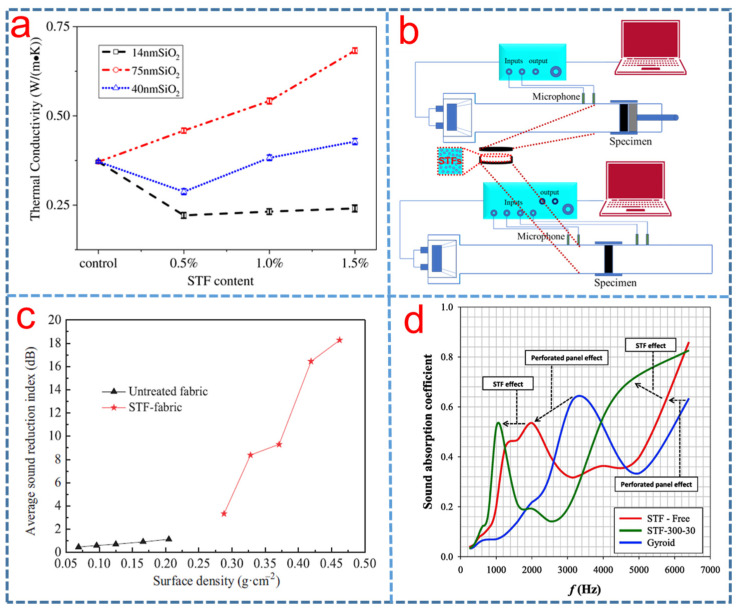
(**a**) Thermal conductivity of the rigid STF/PU foam composites as related to the content of STF and particle size of SiO_2_ [142], (**b**) schematic of the acoustic test experiment setup [7], (**c**) the average acoustic reduction index of samples of different surface density [143] and (**d**) the comparison of sound coefficient measurements of reference configuration with Gyroid and STF-300-30 [58].

**Figure 9 polymers-15-02238-f009:**
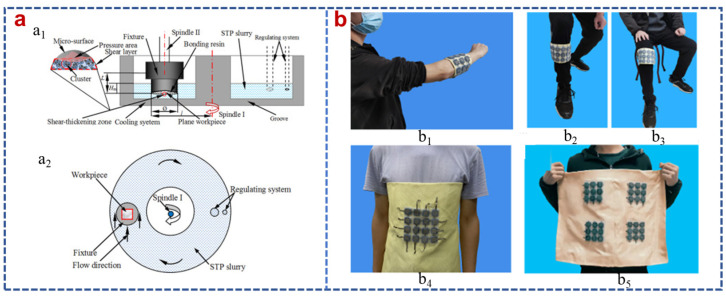
(**a**) Illustration of the polishing test system (**a_1_**) frontal view and (**a_2_**) vertical view [148] and (**b**) the versatile wearable C-STF/Ecoflex/Kevlar ((**b_1_**) worn on arm, (**b_2_**) leg and (**b_3_**) knee, (**b_4_**) the overlying one provided higher protection effect, and (**b_5_**) the arrays were integrated into one pad with larger protection area) [153].

**Table 1 polymers-15-02238-t001:** The compositions of STFs.

Particles	Carrier Fluids	Additives	Reference
Polystyrene-ethylacrylate (PSt-EA)	EG	—	[2]
Polymethyl methacrylate (PMMA)	Glycerine–water	—	[22]
	PEG	—	[23]
Silica nanoparticles	PEG	—	[24]
	Ethyl alcohol and PPG	—	[25]
	PEG	Polyvinyl alcohol	[26]
	Water	—	[27]
	Ionic liquids	—	[28]
	EG	PEG	[29]
	PEG	Graphene	[30]
	Ethanol and PEG	Silane coupling agent	[31]
Fumed silica	PEG	SiC	[32]
		SiC nanowires	[33]
		Carbon nanotubes	[34]
	EG	—	[35]
	PEG	Clay nanoparticles	[29]
Cornstarch	Water	—	[36]
	CsCl in demineralized water	—	[36]
Styrene/acrylate	EG	—	[37]
(Poly)Styrene-acrylonitrile (PSAN)	EG	—	[38]
Polyvinyl chloride (PVC)	Dioctyl phthalate	—	[38]
Precipitated calcium carbonate	PEG	—	[39]
ZrO_2_	Mineral oil	—	[40]
Soda-lime glass spheres	Water	—	[40]
Glass spheres	Mineral oil	—	[40]
Polystyrene (PS)	PEG	—	[41]
Nano-silica and calcium	PEG and ethanol	—	[42]
Kaolin clay particles	Glycerol	—	[43]

## Data Availability

Not applicable.

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
