# Peer review of "Shear Thickening Fluid and Its Application in Impact Protection: A Review"

_polymers, 2023, doi:10.3390/polym15102238_

Round 1
Reviewer 1 Report
Dear Authors
Your review is very comprehensive and provides a good overview of applications of shear thickening fluids. It is well-done concept of organization of the article with suitable sections of applications of STF like ballistic, damping, ad other purposes.
The Introduction part is good but it needs improvement in terms of a better overview of novel references on this topic; many of them a mention in later parts of the manuscript but it should be given a better attraction to readers in the first few paragraphs of the paper where mainly older references are mentioned.
Also, the English should be checked and improved throughout the whole manuscript.
Other parts of particular applications are overviewed with many details and a fine. I hope that authors will find a way to add self-healing applications of these STFs.
The Conclusion is provided in a short concise way, it should be improved (please see comment in the attached version of the article).
The attached document gives comments that could improve the article in a better way. Thank you.

Author Response
Response to Reviewer 1’s Comments
Comment 1: The Introduction part is good but it needs improvement in terms of a better overview of novel references on this topic; many of them a mention in later parts of the manuscript but it should be given a better attraction to readers in the first few paragraphs of the paper where mainly older references are mentioned.
Response: Thanks for the comment. We have added some novel applications and new references in the introduction part as follows:
Line 49: “ Due to the excellent energy absorption and energy dissipation characteristics of STF, it has been widely used in the Li-ion batteries [1] (ref 4 in manuscript), wearable devices [2] (ref 5 in manuscript), triboelectric nanogenerator (TENG) [3] (ref 6 in manuscript), protective structures [4, 5] (ref 7, 8 in manuscript) and some novelty applications [6-8] (ref 9-11 in manuscript). ”
Line 85: “Some approaches to improve the ballistic performance including surface treatments and modifications of fabrics in STF-based soft body armor were elucidated [8, 9] (ref 11, 53 in manuscript). The applications of various STF impregnated fabric composites and the STFs’ contribution on improving the impact, ballistic and stab resistance performance were investigated by many researchers [10, 11] (ref 21, 54 in manuscript).”
Comment 2: Also, the English should be checked and improved throughout the whole manuscript.
Response: The comment is appreciated. We have edited the manuscript carefully to avoid grammar errors.
Comment 3: The Conclusion is provided in a short concise way, it should be improved (please see comment in the attached version of the article.
Response: Thanks for the suggestion. We have revised the conclusion in the manuscript according to the comment as follows:
“This paper provides a comprehensive review of various applications pertaining to STF. There is a growing demand to understand the shear thickening mechanism to explain the mechanical behavior of STF in different applications. Thus, this review discusses five different shear thickening mechanisms, i.e., order-disorder theory, hydro-clustering theory, dilation theory, jamming theory, friction contact theory. Then, this paper reviews typical STF’s applications to provide an insight into the usage of STF in impact protections. First, this paper reviews different methods to improve the impact, ballistic and stab-resistant performance of STF-impregnated fabric. The greater impact resistance and energy absorption of STF-impregnated fabric are mainly attributed to the inherent shear thickening behavior of STF and improvement of the frictions between yarns and fibers. Moreover, the industrial devices based on STFs, such as dampers, shock absorbers and some other novel applications (acoustic structure, STF-TENG and electrospun nonwoven mats), are also included in this review. With the addition of STF, the mechanical, acoustic, thermal, vibration, mechanic-magnetic coupling and electrokinetic performances of devices are improved.
At present, the application of STF to smart wearable devices has become a hot topic of research. However, the sedimentation of STF may restrict its wide application. For the STF, the particle sedimentation of the STF is nearly unavoidable after long-term usage. Therefore, it is recommended to prepare the STF with particles and carrier liquid with comparable densities. Moreover, the shear thickening behavior of STF is also sensitive to the environmental temperature. How to prepare an STF which can work under a wide temperature range is also a meaningful research direction. However, the problems of high viscosity, hygroscopicity and difficult handling have become the biggest factors hindering its development in the practical application of STF. To overcome these shortcomings and to improve the stability of the STF in service, it is particularly important to use suitable technology to encapsulate the STF. The interaction between STF and different encapsulated materials for the impact protection is also a direction for the future research.”
References:
[1] Liu K, Cheng C-F, Zhou L, Zou F, Liang W, Wang M, et al. A shear thickening fluid based impact resistant electrolyte for safe Li-ion batteries. Journal of Power Sources. 2019;423:297-304.
[2] Liu M, Zhang S, Liu S, Cao S, Wang S, Bai L, et al. CNT/STF/Kevlar-based wearable electronic textile with excellent anti-impact and sensing performance. Composites Part A: Applied Science and Manufacturing. 2019;126.
[3] Wang S, Liu S, Zhou J, Li F, Li J, Cao X, et al. Advanced triboelectric nanogenerator with multi-mode energy harvesting and anti-impact properties for smart glove and wearable e-textile. Nano Energy. 2020;78.
[4] Liu H, Fu K, Zhu H, Yang B. The acoustic property and impact behaviour of 3D printed structures filled with shear thickening fluids. Smart Materials and Structures. 2021;31.
[5] Liu H, Zhu H, Fu K, Sun G, Chen Y, Yang B, et al. High-impact resistant hybrid sandwich panel filled with shear thickening fluid. Composite Structures. 2022;284.
[6] Chen S, Zhao Y, Zhang H, Xu P, Jiang Z, Zhang H, et al. Direct microencapsulation of Ionic-Liquid-Based shear thickening fluid via rheological behavior transition for functional applications. Chemical Engineering Journal. 2023;455.
[7] Sheikhi MR, Gürgen S. Anti-impact design of multi-layer composites enhanced by shear thickening fluid. Composite Structures. 2022;279.
[8] Fehrenbach J, Hall E, Gibbon L, Smith T, Amiri A, Ulven C. Impact Resistant Flax Fiber Fabrics Using Shear Thickening Fluid. Journal of Composites Science. 2023;7.
[9] Mawkhlieng, U.; Majumdar, A.; Laha, A. A review of fibrous materials for soft body armour applications. RSC Advances 2020, 10, 1066-1086
[10] Ribeiro, M.P.; da Silveira, P.; de Oliveira Braga, F.; Monteiro, S.N. Fabric Impregnation with Shear Thickening Fluid for Ballistic Armor Polymer Composites: An Updated Overview. Polymers (Basel) 2022, 14.
[11] Xie, Z.; Chen, W.; Liu, Y.; Liu, L.; Zhao, Z.; Luo, G. Design of the ballistic performance of shear thickening fluid (STF) impregnated Kevlar fabric via numerical simulation. Materials & Design 2023, 226.

Reviewer 2 Report
I have reviewed the paper by Liu et al. I found the work to be a comprehensive review of shear thickening fluids and its application to impact protection. I would say that the paper has certain inserted sections (in red) that donot seem to flow with the rest of the manuscript and I would advise the authors to make sure that the sections fit well with the rest of the sections. Example would be lines 62-67 which seem out of place with respect to the following section on categorization of properties.
Author Response
Comment 1: I would say that the paper has certain inserted sections (in red) that donot seem to flow with the rest of the manuscript and I would advise the authors to make sure that the sections fit well with the rest of the sections. Example would be lines 62-67 which seem out of place with respect to the following section on categorization of properties.
Response: Thanks for the comment. We have deleted the lines 62-67 and checked the manuscript carefully to make sure that the inserted sections (in red) fit well with the rest of the sections.
Round 2
Reviewer 1 Report
Dear Authors
Your updated revised version of the manuscript is very much improved and well organized.
It consists of theoretical aspects of STF, as well as their applications for a variety of technical needs. Very good work that overviews the most recent and older literature.
Some minor aspects that could be considered for checking once again and improved your work are given in attached document.
Thanks.

Author Response
Comment 1: Some minor aspects that could be considered for checking once again and improved your work are given in attached document.
Response: Thanks for the comment. We have revised the manuscript as follows:
Line 46: “Shear thickening fluid (STF) is a non-Newtonian fluid that exhibits an abrupt increase in viscosity by a few orders of magnitude with increasing shear rate (1-3).”
Line 50-52: “Due to the excellent energy absorption and energy dissipation characteristics of STF, it has been widely used in the Li-ion batteries (4), wearable devices (5), triboelectric nanogenerator (TENG) (6), protective structures (7,8) and some novel applications (9-11).”
Line 58-60: “The shear thickening mechanisms, material composition, fabrication methods, rheological properties and factors influencing shear thickening behavior have been reviewed by many researchers (15-21).”
Line 562-564: “This paper reviews different methods to improve the impact, ballistic and stab-resistant performance of STF-impregnated fabric.”
Line 572-582: “The particle sedimentation of the STF is nearly unavoidable after long-term usage. Therefore, it is recommended to prepare the STF with particles and carrier liquid with comparable densities. Moreover, the shear thickening behavior of STF is also sensitive to the environmental temperature. How to prepare an STF which can work under a wide temperature range is a meaningful research direction. However, the problems of high viscosity, hygroscopicity and difficult handling have become the biggest factors hindering its development in the practical application of STF. To overcome these shortcomings and to improve the stability of the STF in service, it is particularly important to use suitable technology to encapsulate the STF. Hence, the interaction between STF and different encapsulated materials for the impact protection is a potential field for the future work.”